# Absent phasing of respiratory and locomotor rhythms in running mice

Coralie Hérent[1], Séverine Diem[1], Gilles Fortin[2], Julien Bouvier[1]*

[1]Université Paris-Saclay, CNRS, Institut des Neurosciences Paris-Saclay, Gif-sur-Yvette, France; [2]Institut de Biologie de l'École Normale Supérieure (IBENS), École Normale Supérieure, CNRS, INSERM, PSL Research University, Paris, France

**Abstract** Examining whether and how the rhythms of limb and breathing movements interact is highly informative about the mechanistic origin of hyperpnoea during running exercise. However, studies have failed to reveal regularities. In particular, whether breathing frequency is inherently proportional to limb velocity and imposed by a synchronization of breaths to strides is still unclear. Here, we examined respiratory changes during running in the resourceful mouse model. We show that, for a wide range of trotting speeds on a treadmill, respiratory rate increases to a fixed and stable value irrespective of trotting velocities. Respiratory rate was yet further increased during escape-like running and most particularly at gallop. However, we found no temporal coordination of breaths to strides at any speed, intensity, or gait. Our work thus highlights that exercise hyperpnoea can operate, at least in mice and in the presently examined running regimes, without phasic constraints from limb movements.

## Introduction

The versatile adaptability of breathing to changes in the environment or behavioral state is vital. Probably, the most striking example is the augmentation of ventilation at the transition from rest to running exercise to match the augmented energetic demand (*Bramble and Carrier, 1983*; *Mateika and Duffin, 1995*; *Gariépy et al., 2010*). The hyperpnoea during running is principally supported by an increased respiratory rate which underscores an upregulation of the respiratory rhythm generator in the brainstem (*Del Negro et al., 2018*). While examining the dynamic interactions between respiratory and locomotor movements should inform on the origin and nature of the activatory signal, studies have failed to reveal regularities. Despite this, a common postulate is that respiratory frequency is entrained by that of locomotor movements, through inertial oscillations of the viscera and/or by proprioceptive signals from the limbs impacting the respiratory generator (*Iscoe and Polosa, 1976*; *Bramble and Carrier, 1983*; *Baudinette et al., 1987*; *Alexander, 1993*; *Morin and Viala, 2002*; *Potts et al., 2005*; *Giraudin et al., 2012*).

Two alleged signatures of hyperpnoea to running exercise have in particular fueled this model. Firstly, breathing augmentation during running is often considered to be inherently proportional to the velocity of repetitive limb movements (*Bechbache and Duffin, 1977*; *Eldridge et al., 1981*; *DiMarco et al., 1983*; *Casey et al., 1987*). Yet, opposite findings have also been reported (*Kay et al., 1975*), as well as increased respiratory rates during mental imagery of exercise, that is without actual movements (*Thornton et al., 2001*). Secondly, and this is probably the most controversial aspect, the temporal coordination of breaths to strides (often referred to as the 'locomotor-respiratory coupling' or LRC), is commonly highlighted as a conserved feature of hyperpnoea to exercise (*Bechbache and Duffin, 1977*; *Bramble and Carrier, 1983*; *Alexander, 1993*; *Corio et al., 1993*; *Mateika and Duffin, 1995*; *Lafortuna et al., 1996*; *Boggs, 2002*). However, studies in human participants reported a strong heterogeneity in LRC between individuals, from a constant degree of coupling to no coupling at all (*Kay et al., 1975*; *Bernasconi and Kohl, 1993*; *Daley et al., 2013*;

*For correspondence:
julien.bouvier@cnrs.fr

**Competing interests:** The authors declare that no competing interests exist.

*Stickford et al., 2015*). The LRC may also be favored by auditory cues (*Bernasconi and Kohl, 1993*) or by experience (*Bramble and Carrier, 1983*), arguing for the contribution of multiple factors including external stimuli and training. In quadrupeds, the fewer studies available, essentially on running performant species including rabbits, dogs, cats, and horses (*Bramble and Carrier, 1983*; *DiMarco et al., 1983*; *Corio et al., 1993*; *Lafortuna et al., 1996*), again revealed various degrees and ratios of locomotor-respiratory coordination, and raised the possibility that faster running gaits (i.e. gallop) may impose a stronger coupling. Therefore, a major source of confound about the coordination of locomotor and respiratory rhythms may owe to the variety of species examined thus far, and to the attendant variability in ambulatory modes and in the contributions of pre-determined (i.e. hardwired) versus secondary (i.e. sensory, volitional, acquired through experience) factors. Another source of confound probably lies in the distinct indicators used across studies to report the coordination between respiratory and locomotor rhythms (see Discussion). Therefore, while the above respiratory-locomotor interactions *can* occur, it is thus far from clear that they are an obligatory feature of hyperpnoea to exercise and are the manifestation of pre-determined circuits that impose on respiratory frequency that of the locomotor movements.

Laboratory mice have become the premier model for investigating complex integrated tasks including adaptive locomotor and respiratory control (*Benarroch, 2007*; *Bouvier et al., 2010*; *Ramanantsoa et al., 2011*; *Talpalar et al., 2013*; *Bouvier et al., 2015*; *Ruffault et al., 2015*; *Kiehn, 2016*; *Del Negro et al., 2018*; *Usseglio et al., 2020*) with promising benefits for human health (*Amiel et al., 2003*; *Benarroch et al., 2003*; *Lavezzi and Matturri, 2008*). Mice also stand as a good model for investigating the physiological basis and benefits of running exercise (*Lerman et al., 2002*; *Lancel et al., 2003*; *Sartori et al., 2020*). Furthermore, by being housed and raised in a standardized manner across laboratories, mice should give access to the hardwired manifestation of hyperpnoea to exercise, that is with minimal influence of volitional control, variations in external stimuli, or prior experience. Finally, mice benefit from a large array of 'Omics' toolboxes that allow the manipulation of signals, cell types and neural circuit activity combined to quantitative measurements of behaviors. However, the dynamics of hyperpnoea to exercise had yet to be characterized in this resourceful species, to pave the way for hypothesis-driven investigations of its cellular and circuit underpinnings. To this aim, we developed a novel method for monitoring inspiratory activity chronically, which we combined with video-tracking of the four limbs during unrestrained running on a motorized treadmill operating at different speeds. We reveal that inspiratory frequency augments during trotting by about twofolds to a fixed and stable set point value, irrespective of trotting velocities and of inclination. Yet, respiratory rate was further enhanced during escape running, and most significantly at gallop. We also demonstrate the absence at all of temporal coordination of breaths to strides at any speed, intensity, or gait. Our work therefore highlights a hardwired mechanism that discretely sets respiratory frequency independently of limb movements but in line with the engaged locomotor program.

## Results

### Chronic electromyographic recordings of diaphragmatic activity in running mice

Examining whether and how the rhythms of limb and breathing movements interact in mice requires to access breathing parameters in freely moving conditions. As an alternative to measuring ventilation by whole body plethysmography (WBP [*DeLorme and Moss, 2002*]) which is hardly compatible with displacement movements, or to the constraining use of air-tight chambers (*Tsuchiya et al., 2012*), we extended to mice the use of electromyography (EMG) recordings of the diaphragm (*Fat-Chun Tony Chang and Harper, 1989*; *Shafford et al., 2006*), the main inspiratory muscle. For this, we placed two steel wires within the peritoneum to superficially contact the diaphragm, that is without passing through the muscle itself (*Figure 1A,B*; see Materials and methods for details). Upon full recovery, we first placed EMG-implanted animals in a customized WBP chamber and confirmed that diaphragmatic neurograms correlate with the rising phase (i.e. inspiration) of the plethysmography signal, an estimate of the respiratory volume (*Figure 1C*). We used the alternating phases of activity and inactivity of diaphragmatic EMG to define respectively inspiration and expiration and to measure their durations (inspiratory time Ti; expiratory time Te; *Figure 1D*). The cycle-to-cycle interval

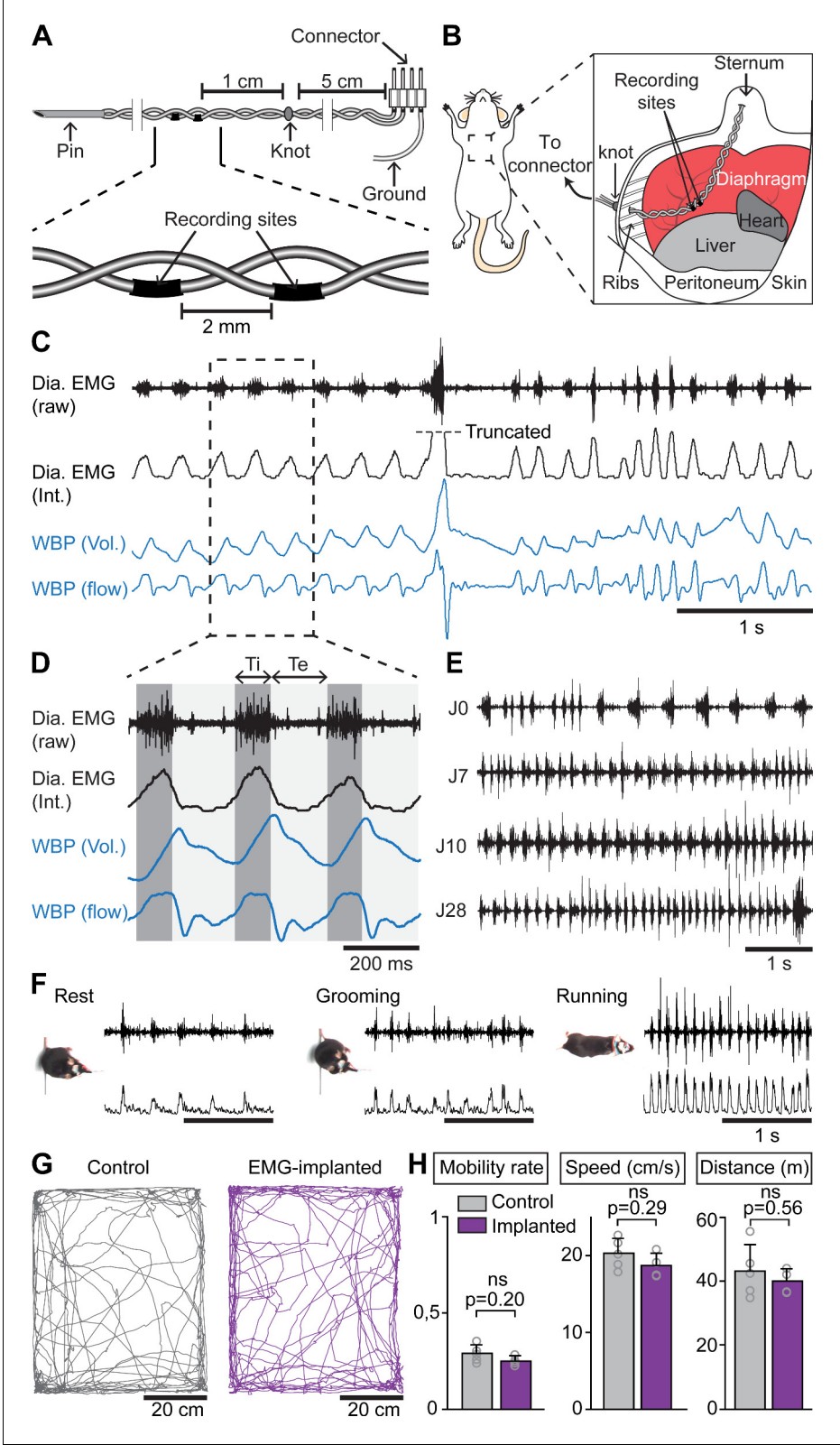

**Figure 1.** Chronic electromyographic recordings of the diaphragm to monitor inspiration in freely moving mice. (A, B) Schematics of the EMG recording electrodes and of the diaphragm implantation. (C) Simultaneous diaphragm EMG recordings (raw and integrated traces in black) and whole-body plethysmography (WBP, volume and flow, in blue) showing that the diaphragm neurogram estimates inspiratory flow. (D) Enlarged view of three

*Figure 1 continued on next page*

*Figure 1 continued*

inspiratory bursts highlighting the following respiratory parameters: inspiratory time (Ti) defined as a bout of diaphragm activity, and expiratory time (Te) as the silent period between bouts. (**E**) Diaphragm activity recorded on the day of the surgery (J0) and at 7, 10, and 28 days post-surgery (representative of four mice). (**F**) Raw and integrated diaphragm neurograms in an open field test during rest, grooming and spontaneous running showing preserved recording quality in spite of movements. (**G**) Locomotor trajectories of one representative non-implanted mouse (gray) and one representative EMG-implanted mouse (purple) for 10 min in the open field. (**H**) Bar-graphs showing the mean ± SD mobility rate, locomotor speed during mobility and total distance traveled in control (n = 5) and EMG-implanted mice (n = 4). p values in (**H**) are obtained from Mann-Whitney *U* tests and considered not significant when p>0.05.

The online version of this article includes the following source data for figure 1:

**Source data 1.** Locomotor parameters in the open field.

---

(Ti+Te) was used to obtain the instantaneous frequency of each inspiratory burst, leading to respiratory rate. The peak amplitude of integrated EMG signals appeared as an estimate of inspiratory flow obtained as the first derivative of plethysmographic signal (*Figure 1C,D*). Additionally, these measurements could be repeated over up to 28 days post-surgery without signal degradation (*Figure 1E*). Importantly, and in contrast to WBP, our EMG recordings provide stable inspiratory activity independently of the animal's displacement movements and can therefore be performed in freely behaving mice (*Figure 1F*). Finally, implanted and connected animals behaved similarly as non-operated mice in an open field arena (*Figure 1G,H*) indicating that the implantation did not alter the animal's ability to move spontaneously nor induced pain.

## Breathing frequency augments independently of displacement speed during treadmill trotting

The most-common gait used by mice for running is the trot, characterized by simultaneous forward movements of the diagonal limb pairs and alternation of homologous limb pairs (*Bellardita and Kiehn, 2015*; *Lemieux et al., 2016*). At trot, variations in step frequency allow to cover a wide range of displacement speeds (*Bellardita and Kiehn, 2015*; *Mayer et al., 2018*). To examine how breathing frequency changes at different regimes of trot, we placed EMG-implanted animals on the belt of a motorized treadmill. We selected four representative speeds (15, 25, 40, and 50 cm/s) that cover most of the range achieved at trot (*Bellardita and Kiehn, 2015*; *Lemieux et al., 2016*) and are accessible without prior training or aversive motivation (*Fernando et al., 1993*). Displacement speeds below 15 cm/s were not analyzed since they are typically manifested by intermittent bouts of walking interleaved with non-exercise behaviors, notably whisking and sniffing that also mobilize the respiratory apparatus and would be a source of confound.

When the animals engaged in stable running, that is their displacement speed was in phase with that of the treadmill, their respiratory rate (collected during the first 1.5 min) was increased but, surprisingly, independently of the running speed. Indeed, respiratory frequency increased by 245% at 15 cm/s, *225% at 25 cm/s, 233% at 40 cm/s*, and by *226%* at 50 cm/s, without significant difference between the four speeds (*Figure 2A–B*). The increase in respiratory frequency from baseline was associated with a decrease of both Ti and Te (*Figure 2C,D*). Additionally, the amplitude of integrated EMG bursts was significantly higher during running than rest, but still without clear relation with speed (*Figure 2E*). *To examine the impact of a longer run*, and exclude an effect of acute stress during the first minutes after starting the treadmill, *animals were challenged* to a continuous 10-min run at 40 cm/s, the highest speed maintainable without prior training. We found the respiratory frequency to be stable throughout the period (*Figure 2—figure supplement 1*). *Next*, to examine the impact of workload onto breathing changes, mice were submitted to the same running speeds on a treadmill with a 10% incline (*Gardiner et al., 1982*; *Gillis and Biewener, 2002*, *Figure 2F–J*). We found that changes in respiratory parameters were overall similar to the values obtained on the level treadmill and were still independent of running speed (*Figure 2G–J*). These experiments altogether show that, during trotting on a treadmill in mice, respiratory rate increases to a fixed set point value that is independent of the animal's displacement speed. Furthermore, running on a 10% inclined treadmill did not further enhance breathing rate, suggesting little influence of the exercise grade.

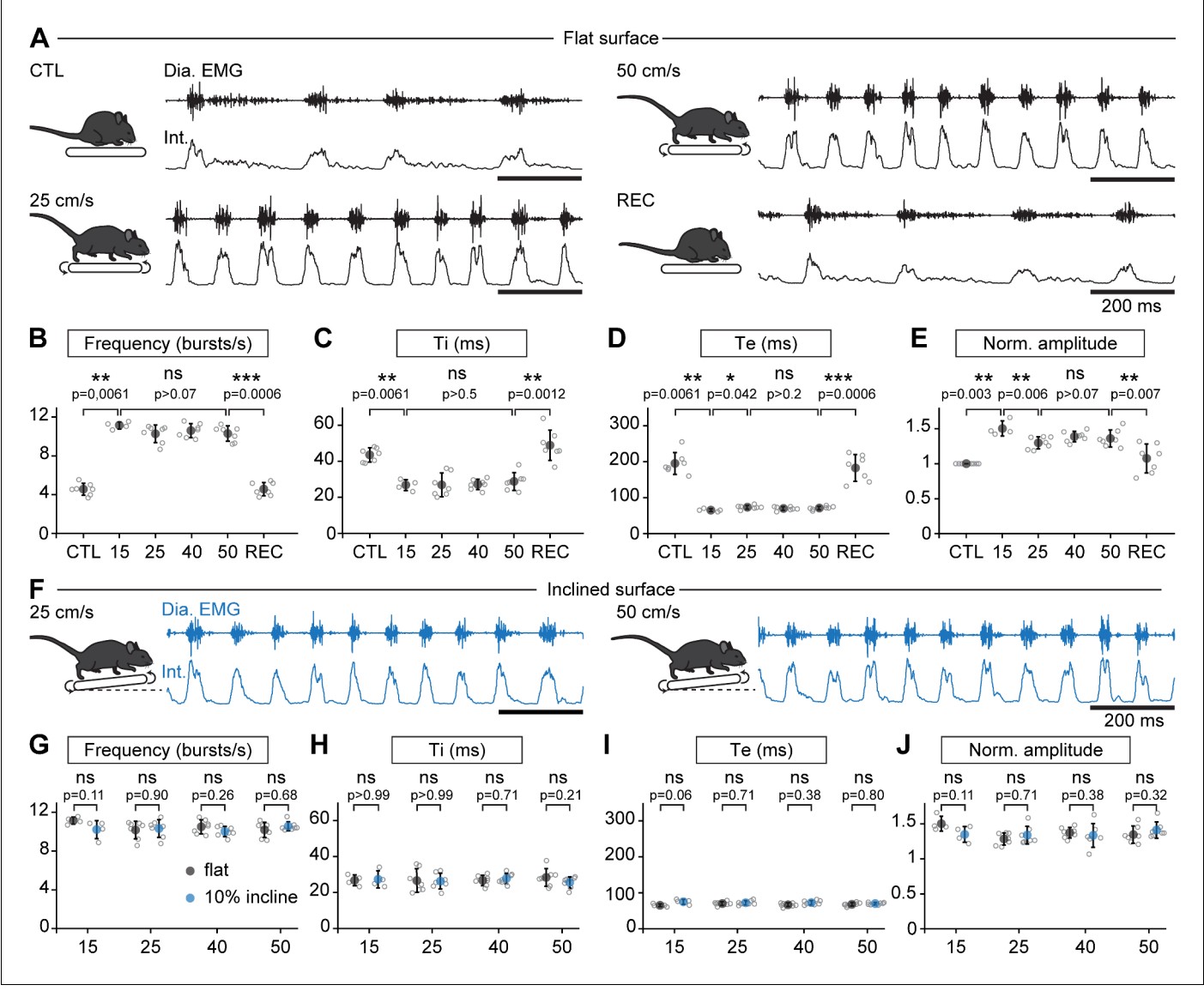

**Figure 2.** Breathing rate augments during trotting independently of limb velocity or surface inclination. (**A**) Diaphragm activity recordings in control condition (CTL), trotting at 25 and 50 cm/s and during recovery after the run (REC). Raw (Dia. EMG) and integrated (Int.) signals are illustrated in each condition. (**B–E**) Analysis of respiratory parameters in control condition, and during trot at 15, 25, 40, and 50 cm/s, and during recovery: frequency (**B**), inspiratory (Ti, **C**) and expiratory (Te, **D**) times and normalized amplitude (**E**). Data are mean ± SD from n = 7 mice per condition for all speeds except at 15 cm/s (n = 4) where three animals had unstable running and were discarded. p values indicated are obtained from Mann-Whitney *U* tests and considered not significant (ns) when p>0.05. (**F**) Raw (Dia. EMG) and integrated (Int.) diaphragmatic activity recordings during trot at 25 and 50 cm/s on a treadmill with 10% incline. (**G–J**) Similar analyses as in (**B–E**) during running on the inclined treadmill (blue) compared to the values on the flat treadmill (black). p values indicated are obtained from Mann-Whitney *U* tests and considered not significant when p>0.05. See *Figure 2—figure supplement 1* for a similar analysis during a 10-min run.

The online version of this article includes the following source data and figure supplement(s) for figure 2:

**Source data 1.** Respiratory parameters on the flat (B–E) and inclined (G–J) treadmill.
**Figure supplement 1.** Respiratory changes during a prolonged treadmill running.
**Figure supplement 1—source data 1.** Respiratory parameters (panels B, C).

## Breaths are not temporally synchronized to strides during treadmill trotting

While our observations above suggest that the respiratory rhythm can operate without constraints from locomotor movements, a temporal coordination between breaths and strides could occur at

specific regimes (*Lafortuna et al., 1996*). To examine this, each limb was tracked using DeepLabCut (*Mathis et al., 2018*; *Mathis and Mathis, 2020*, *Figure 3A–B*, see Materials and methods) and their coordinates were used to register the time of footfall as well as the stance and the swing phases that define the locomotor cycle (*Bellardita and Kiehn, 2015*, *Figure 3C*). We then expressed the onset of individual inspiratory bursts within the locomotor cycle as a phase value (ΦInsp) from 0 (preceding footfall, $FF_{n-1}$) to 1 (footfall, $FF_n$, *Figure 3C*). We then reported ΦInsp values of at least 35 consecutive breaths for one representative animal, as well as the mean concentration of phases for each animal, on a circular diagram (*Kjaerulff and Kiehn, 1996*, see Materials and methods). During trotting on the level treadmill, we found that inspiratory bursts could occur at any moment of the locomotor cycle, regardless of the limb considered as a reference. Indeed, ΦInsp values were evenly distributed across a circular plot diagram (*Figure 3D*, black marks on the outer circle are for one representative animal) resulting in non-oriented mean phase values (colored dots within the inner circle, one color dot per animal). Consequently, ΦInsp values collected from all animals were evenly distributed across the entire locomotor cycle (*Figure 3D*, bar-graphs). These observations were consistent at both the lowest (15 cm/s), intermediate (25 cm/s) and fastest (50 cm/s) trotting speeds accessible on the treadmill (*Figure 3D*), as well as with a 10% incline (*Figure 3E*). We found similar results when mice were challenged to a 10-min run (*Figure 3—figure supplement 1*). Furthermore, correlating diaphragmatic activity with either the forelimb or hindlimb resulted in a flat cross-correlogram (*Figure 3—figure supplement 2*), confirming the absence of common modulation between breaths and strides, even at a higher ratio. Altogether, these data demonstrate that respiratory rate increases without any phasing of breaths to locomotor movements, at least at the most common regimes of trot accessible without training and aversive conditioning on treadmill.

## Breathes are not temporally correlated to strides at higher displacement speeds, including at gallop, in a linear corridor

While the conditions examined above on the treadmill cover a large range of the displacement speeds in mice, these animals can engage into even faster regimes (*Bellardita and Kiehn, 2015*; *Lemieux et al., 2016*). Furthermore, at these most demanding displacement speeds, mice can, like other quadrupeds, increase step frequency further using a galloping gait defined by synchronized movements of the left and right hindlimbs and alternating movements of the left and right forelimbs (*Heglund and Taylor, 1988*; *Bellardita and Kiehn, 2015*; *Caggiano et al., 2018*; *Josset et al., 2018*). We hence investigated how breathing rate adjusts at these faster running regimes, including gallop, in mice. To be engaged in these regimes without resorting to aversive to conditioning on a treadmill, EMG-implanted animals were placed in a linear corridor and a brief air puff was applied to the back of the animal (*Caggiano et al., 2018*). This could engage the animals in either of few bouts of trot, at an average displacement speed of 72 ± 9 cm/s (*Figure 4A,B*), or in a few bouts of gallop at 98 ± 11 cm/s (*Figure 4D,E*). When trot was evoked, respiratory frequency increased by 247% from baseline, to a value slightly higher than at 50 cm/s on the treadmill (*Figure 4G*). However, analyzing the temporal coupling of breathes to strides using the same methods as above revealed that breath onsets were again evenly distributed across the locomotor cycle regardless of the limb considered (*Figure 4C*). When the air puff instead evoked a few bouts of gallop (*Figure 4D,E*), we found that the respiratory frequency was increased by 314% from baseline to a value significantly higher than that measured during trot in both the treadmill and the corridor (*Figure 4G*). This respiratory frequency increase owed to a further decrease in Ti and Te (*Figure 4H,I*), and the amplitude of inspiratory bursts was also increased compared to trotting values on the treadmill (*Figure 4J*).

During gallop, the 'hopping-like' movements of the hindlimbs impose a stronger constraint on the visceral mass that may act as a piston mechanism (*Baudinette et al., 1987*; *Alexander, 1993*; *Bramble and Jenkins, 1993*) assisting in producing respiratory airflow. Therefore a constrained occurrence of diaphragmatic movements to a specific phase of the locomotor cycle may be preferred, since possibly most advantageous, at this gait (*Baudinette et al., 1987*; *Lafortuna et al., 1996*; *Boggs, 2002*). We thus examined whether gallop favored a temporal coupling of breaths to strides in mice using the same analytic methods as for trot. Our data however show that, similar to the trot conditions, breath onsets during gallop were evenly distributed across the locomotor cycle, whether defined using the alternating forelimbs or the left-right synchronized hindlimbs (*Figure 4E, F*).

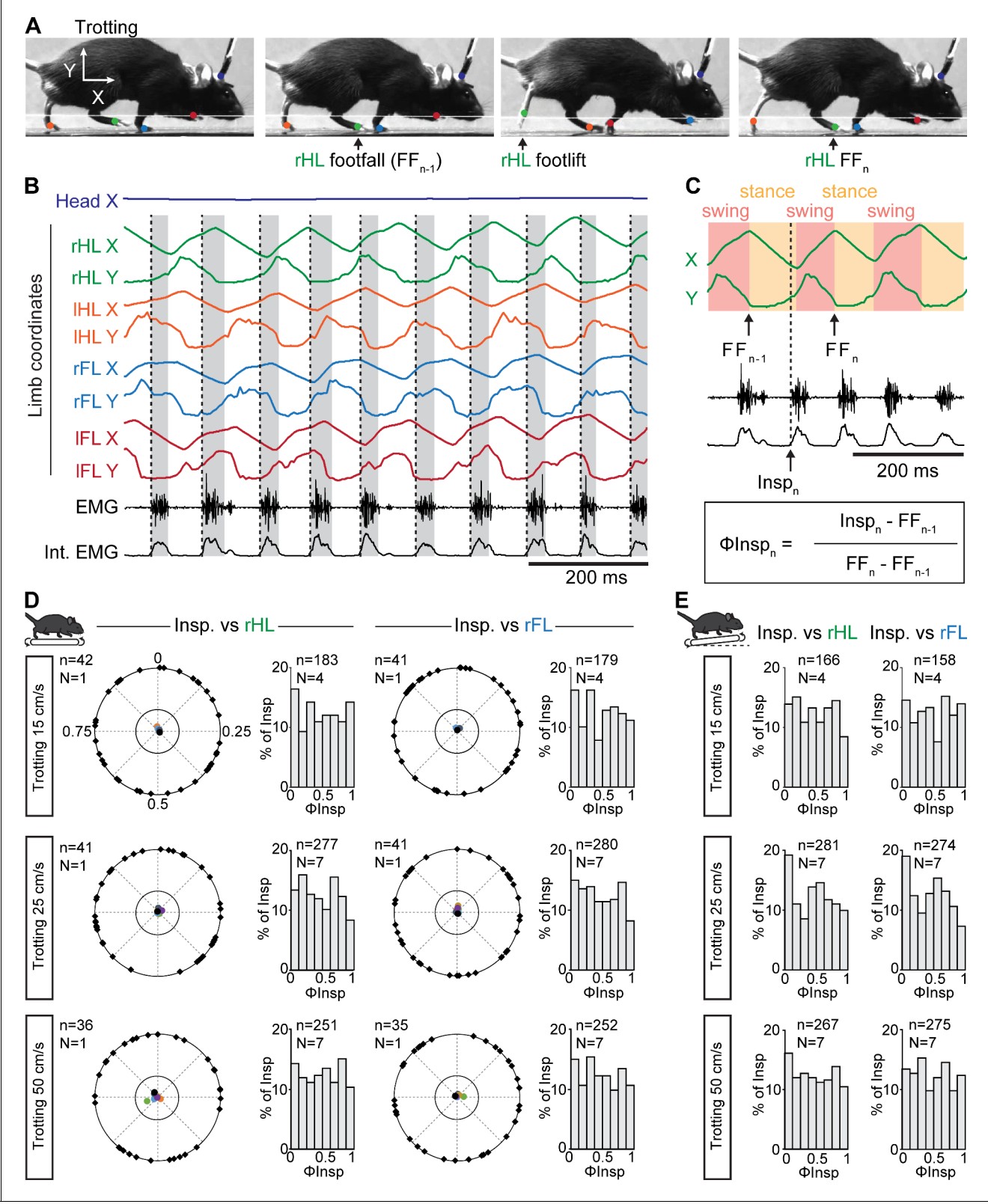

**Figure 3.** Breaths are not temporally-synchronized to strides during trotting on a treadmill. (**A**) Side views of one representative trotting mouse on a flat treadmill. The four limbs and the head were tracked and color-labeled: right hindlimb (rHL, green), left hindlimb (lHL, orange), right forelimb (rFL, blue), left forelimb (lFL, red) and head (dark blue). One complete rHL locomotor cycle is shown, between two consecutive footfalls ($FF_{n-1}$ and $FF_n$). (**B**) Horizontal (X) and vertical (Y) coordinates of the tracked limbs as well as raw (EMG) and integrated (Int. EMG) diaphragmatic neurograms, during stable

*Figure 3 continued on next page*

*Figure 3 continued*

trot. Dotted black lines indicate the onsets of inspiratory bursts and shaded rectangles highlight inspiratory times. (C) Enlarged view of locomotor cycles of the rHL showing the swing and stance phases. The occurrences of inspiratory bursts (Insp$_n$) within the locomotor cycle are expressed as a phase value ($\Phi$Insp$_n$) from 0 (FF$_{n-1}$) to 1 (FF$_n$). (D) Circular plots diagrams showing the phase-relationship between individual inspiratory bursts and the indicated reference limb for one representative animal trotting at 15, 25, and 50 cm/s on a flat treadmill. Black diamonds on the outer circle indicate the phase of n individual inspirations. The black dot indicates the mean orientation vector for that animal and the colored dots indicate the mean orientation vector of 3 (at 15 cm/s) or 6 (at 25 and 50 cm/s) other animals. The positioning of these mean values within the inner circle illustrates the absence of a significantly oriented phase preference (R < 0.3, with R being the concentration of phase values around the mean as defined in *Kjaerulff and Kiehn, 1996*). Bar-graphs to the right are distribution histograms of the phases of inspiratory bursts and the same reference limb for all n events from N animals. (E) Phase distribution histograms between inspiratory bursts and the indicated reference limb for all n events from N animals running at 15, 25, or 50 cm/s on the inclined treadmill. Note that inspiratory bursts in (D) and (E) are evenly distributed across the entire locomotor cycle in each condition and that distribution histograms do not show a phase-preference. See also *Figure 3—figure supplement 1* for a similar analysis during a 10-min run, and *Figure 3—figure supplement 2* for cross-correlograms between inspiratory and limb activity.

The online version of this article includes the following source data and figure supplement(s) for figure 3:

**Source data 1.** Circular plots for the flat treadmill conditions (panel D).
**Source data 2.** Distribution histograms for both the flat and inclined treadmill (panels D and E).
**Figure supplement 1.** Correlation of breaths to strides during prolonged treadmill running.
**Figure supplement 1—source data 1.** Circular plots.
**Figure supplement 1—source data 2.** Distribution histograms.
**Figure supplement 2.** Example cross-correlations of breathing and locomotor rhythms showing no common modulation.

Altogether, these results indicate that the respiratory frequency increase is more pronounced when mice engage in faster running regimes associated with escape, and particularly at gallop. However, even at these regimes, breaths are not temporally coordinated to strides.

## Prior training does not favor a temporal synchronization of breaths to strides

Since the coordination between breathing and stride rhythms may increase as a function of experience level (*Bramble and Carrier, 1983*), we next reasoned that some degree of locomotor respiratory synchronization may be acquired through training. Therefore, three animals underwent a 8 week training consisting in (1) free access to a running wheel in the cage, a paradigm that suffices to have multiple benefits typically associated with exercise, including enhanced locomotor learning and skills, changes in synaptic and axonal function and anti-inflammatory actions (*Lancel et al., 2003*; *Parachikova et al., 2008*; *Li and Spitzer, 2020*) and (2) daily training on the treadmill, also shown to improve motor skill learning through enhanced synaptic and axonal function (*Chen et al., 2019*, *Figure 5*). Animals were then implanted with EMG electrodes and challenged to treadmill running on a level or inclined treadmill as well as to the air puff driven gallop as performed above. We observed similar changes in breathing frequency and burst amplitude as in untrained mice (data not shown), and importantly, we found that the onsets of inspiratory bursts were still evenly distributed across the locomotor cycle with no phase preference for any animal and in any condition (*Figure 5B–D*). Altogether, these experiments and cycle-to-cycle correlation analyses demonstrate that respiratory frequency in mice increases without breaths being temporally synchronized to locomotor movements, regardless of locomotor speed, grade, gait, or prior training.

## Discussion

We provide here the first examination of breathing changes and of the coordination of breaths with strides during running in the resourceful mouse model. This was made possible by a unique method for implanting EMG electrodes on the diaphragm combined with limb video tracking. Contrary to WBP, this allows repetitive, long duration and artifact-free recordings of inspiratory activity during locomotor displacements. Furthermore, our method eliminates the need to use an air-tight chamber for measuring breathing during exercise (*Tsuchiya et al., 2012*), and is thus compatible with a variety of laboratory environments including large open fields (*Figure 1*), standard treadmills (*Figures 2* and *3*), and corridors (*Figure 4*). Our method was designed to specifically monitor inspiratory activity, the only breathing phase (i) maintained throughout the rest/activity cycle (expiration is passive at

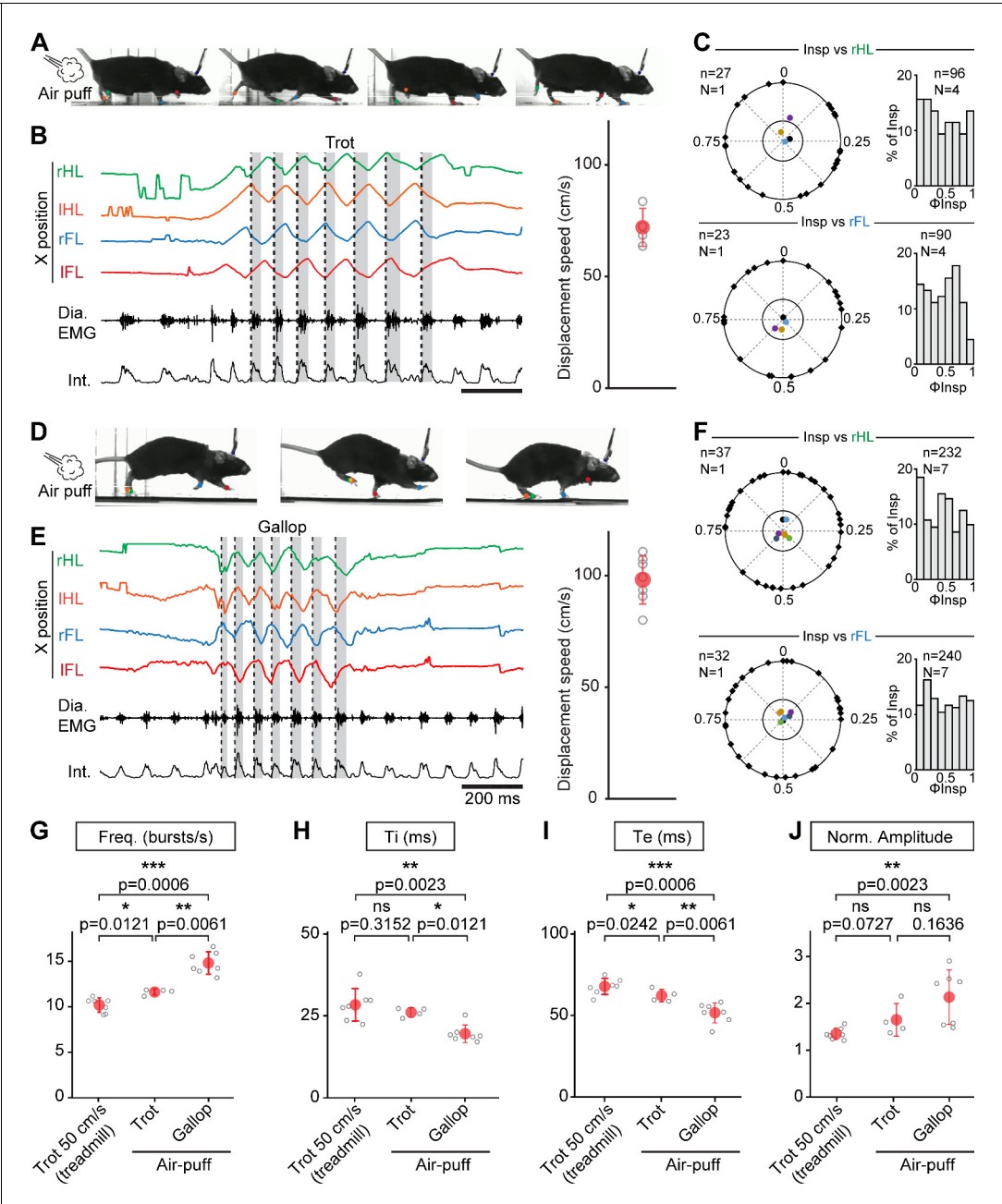

**Figure 4.** Further augmentation of breathing rate yet without temporal correlation of breaths to strides during escape running, including at gallop. (**A**) Side views of one representative mouse during air puff induced trot where right and left hindlimbs (rHL, green; lHL, orange) and forelimbs (rFL, blue; lFL, red) are alternating. (**B**) Changes in the horizontal (X) axis of the limbs over time as well as raw (Dia. EMG) and integrated (Int.) diaphragmatic neurograms are shown before and during four cycles of trot induced by air puff. Dotted black lines indicate the onsets of inspiratory bursts and shaded areas highlight inspiratory times. On the limb coordinates, rapid deflections before the onset of movement are occasional failures in detecting the paws at rest. The graph to the right shows the average running speed of each animal (gray open circles) and the mean ± SD across animals (in red) during air puff induced trot. (**C**) Circular plots showing the phase-relationship between n inspiratory bursts and the indicated reference limb for one representative animal (black diamonds on the outer circle). The black dot indicates the mean orientation vector for that animal and the colored dots indicate the mean orientation vector for three other animals. The positioning of these mean values within or close to the inner circle illustrates the absence of a significantly oriented phase preference (R < 0.3, with R being the concentration of phase values around the mean as defined in *Kjaerulff and Kiehn, 1996*). Bar-graphs to the right are distribution histograms of the phases of inspiratory bursts and the same reference limb for all n events from N = 4 animals. (**D–F**) Same representation as (**A–C**) during air puff induced gallop where right and left hindlimbs are now synchronized and right and left forelimbs are alternating. The average displacement speed of the animals at gallop (98 ± 11 cm/s) is significantly higher than at trot (72 ± 8 cm/s, p=0.0121, Mann-Whitney *U* test). Circular plots and distribution histograms are obtained from N = 7 animals. (**G–J**) Changes in respiratory frequency

*Figure 4 continued on next page*

*Figure 4 continued*

(G), inspiratory (Ti, H) and expiratory (Te, I) times and normalized amplitude (J) between trot at 50 cm/s on the treadmill, and air puff induced trot and gallop. Note that breathing rate is most significantly increased at gallop. Data are presented as mean ± SD (trot treadmill: N = 7 mice; trot air puff: N = 4 mice; gallop air puff N = 7 mice) and p values indicated are obtained from Mann-Whitney *U* tests.

The online version of this article includes the following source data for figure 4:

**Source data 1.** Circular plots (panels C and F).
**Source data 2.** Distribution histograms (panels C and F).
**Source data 3.** Respiratory parameters (panels G–J).

rest) and (ii) with a known pacing origin - the preBötzinger complex, the main respiratory rhythm generator (*Ausborn et al., 2018*; *Del Negro et al., 2018*). Therefore, diaphragm EMGs constitute an accurate readout of the temporal organization of sequences of neuronal activity in executive inspiratory circuits. Compared to implanted nasal probes (*Kurnikova et al., 2017*) our EMG approach allows a sampling of respiratory activity with much higher temporal resolution. It thus constitutes a timely addition to the toolbox for studying the adaptive control of breathing.

The dynamics of respiratory changes observed during acute running exercise in laboratory mice, housed and raised in a standardized manner across laboratories, make it possible to draw up an outline of the hardwired interactions between respiration and locomotion, that is with minimal contribution of conditioning or prior experience. Through that, it will help future hypothesis-driven attempts to identify the long-sought physiological, likely neuronal, substrate for hyperpnoea to exercise (*Mateika and Duffin, 1995*; *Gariépy et al., 2010*; *Paterson, 2014*). On that matter, one major finding of our work is the absence at all of a temporal locking of breaths to strides in both naive and trained mice. These findings were first observed during treadmill running up to 50 cm/s, which represent the comfortable speed of volitional running in this species. As previously reported (*Fernando et al., 1993*), we could not engage animals at higher running speeds on the treadmill without prior training with electric shocks, but we also investigated the coordination of breaths to strides during over-ground running in a linear corridor. There, animals engage to higher displacement speeds at trot (72 cm/s on average) and utilize gallop for the fastest displacements (98 cm/s), as previously reported (*Bellardita and Kiehn, 2015*; *Lemieux et al., 2016*). Yet, no breath-to-stride temporal locking was seen at these regimes either. Although we may not have covered the most extreme speeds in this species (*Lemieux et al., 2016*), our work argues that a synchronization of breaths to strides is not a typical feature of respiratory hyperpnoea during running in mice. This lack of coordination should be discussed thoroughly since its existence has been alleged in all species. It must be stressed, however, that the indicator that is most typically used, the 'locomotor respiratory coupling' (or LRC), has a very ambiguous interpretation. In many instances, it only reports the frequency ratio between the two movements, that is locomotor events per respiratory events (*Bechbache and Duffin, 1977*; *Bramble and Carrier, 1983*; *Paterson et al., 1986*; *Corio et al., 1993*). There, the terms 1:1 'coupling' or 'entrainment' therefore refer to the locomotor frequency being, on average, not statistically different to that of breathing. It however says nothing about the phasing of both movements on a cycle-to-cycle basis as performed here. Closest studies are those measuring the 'LRC ratio', the 'degree of LRC', or the 'LRC %', that is the percentage of inspirations starting in the same phase of the step cycle. The distinction between the two measurements is clearly illustrated in the investigation of *Lafortuna et al., 1996*. This study reported that at all running speeds, horses adopt a breathing frequency that is systematically equal to that of the limbs. Yet, the temporal locking of breathes to strides is only intermittent at trot and becomes more frequent at gallop. Similarly in running cats, respiratory frequency is half that of locomotion but with only little common modulation between diaphragmatic and quadriceps activities (*Iscoe, 1981*). Pertinent breaths-to-strides analyses are more common in humans but reveal a strong heterogeneity between individuals, from a constant degree of cycle-to-cycle coupling to no coupling at all (*Kay et al., 1975*; *Bernasconi and Kohl, 1993*; *Daley et al., 2013*; *Stickford et al., 2015*). Therefore, it is often far from clear, especially in animal studies, to which of the above parameters (ratio of frequencies or actual synchronous events) the typical 1:1, or 2:1 'coupling' or 'entrainment' refers to. Making the distinction is yet important when thinking on the neuronal substrates of exercise hyperpnoea. For a temporal locking of breaths and strides, a synchronizing neural command needs to be

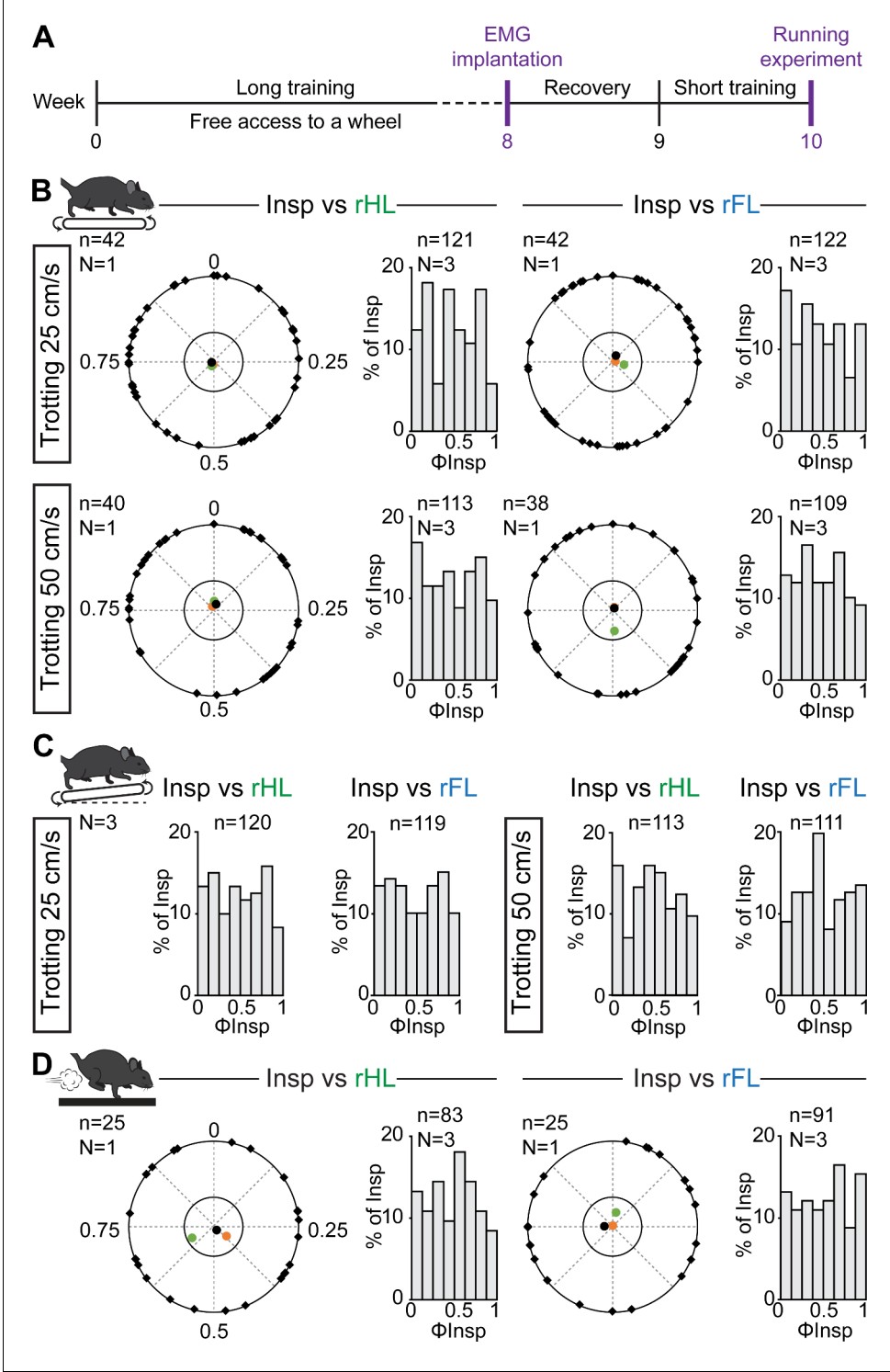

**Figure 5.** Respiratory changes during running in trained mice and phase-relationship analysis between breaths and strides. (**A**) Experimental timeline. Mice were trained for 8 weeks before being implanted for EMG recordings of the diaphragm. (**B**) Circular plots showing the phase-relationship between inspiratory bursts and the indicated reference limb for one representative animal trotting at 25 (top) and 50 cm/s (bottom). Black diamonds on the outer circle indicate the phase of n individual inspirations. The black dot indicates the mean orientation vector for that animal and the colored dots the mean orientation vector of two other animals. The positioning of these mean values within the inner circle illustrates the absence of a significantly-oriented phase preference (R < 0.3, with R being the concentration of phase values around the mean as defined in *Kjaerulff and Kiehn, 1996*). Bar-graphs to

*Figure 5 continued on next page*

*Figure 5 continued*

the right are distribution histograms of the phases of inspiratory bursts and the same reference limb for all n events from three animals. (**C**) Phase distribution histograms between inspiratory bursts and the indicated reference limb for all n events from three animals running at 25 (left) or 50 cm/s (right) on the inclined treadmill. (**D**) Circular plot showing the phase-relationship between inspiratory bursts and the indicated limb for one representative trained animal during gallop. Black diamonds on the outer circle indicate the phase of n individual inspirations. The black circle indicates the mean orientation vector for that animal and the colored circles the mean orientation vector of two other animals. To the right are phase distribution histograms between inspiratory bursts and the same reference limb for all n events from three animals. Note that all inspiratory bursts are evenly distributed across the entire locomotor cycle in all conditions.

The online version of this article includes the following source data for figure 5:

**Source data 1.** Circular plots (panels B and D).
**Source data 2.** Distribution histograms (panels B–D).

---

produced from either active respiratory or the locomotor networks and be fed onto respectively the locomotor or respiratory networks. Admittedly, the complete absence of synchronization of breaths to strides we report here in selected contexts in the C57BL/6J mouse strain does not ascertain that synchronization *never* occurs in mice. One should notably acknowledge that exercise capacity varies among mouse strains, and in between individuals within a strain (*Lerman et al., 2002*; *Meek et al., 2009*). The C57BL/6J line used here having relatively low endurance exercise capacity (*Avila et al., 2017*) it would be interesting to know if FVB/NJ and SWR/J mice with high endurance exercise capacity present comparable, or distinct, adaptive variations. It would also be very informative to examine the impact of longer running sessions, and/or more intense training, for instance until exhaustion (*Avila et al., 2017*) and during sustained gallop. That being said, our findings in multiple running contexts, and the ambiguities raised above from other species make it unlikely that breaths/strides synchronization constitutes a hardwired and obligatory component of respiratory adaptation to running exercise in quadrupeds. Therefore, the implication of phasic signals (i.e. from sensory feedbacks or visceral oscillations) may be a complementary, rather than obligatory, mechanism of exercise hyperpnoea. This possibility is also supported by the persistence of a normal ventilatory response to exercise following the removal of peripheral signals in animal models (*Eldridge et al., 1981*; *Eldridge et al., 1985*) and human subjects (*Fernandes et al., 1990*).

Our data instead favors the quest, at least in mice and during slow-to-moderate trotting speeds on the treadmill, of a primary mechanism that sets respiratory frequency independently of the locomotor cycle and even independently of the velocity of limb movements. Indeed, at running speeds between 15 and 50 cm/s, respiratory rate increased in a step-like manner by about twofolds from rest, and was stably maintained thereafter throughout the running period, irrespective of trotting velocities and regardless of difference in height of the route. This is reminiscent of the respiratory changes previously documented during swimming in the lamprey (*Gravel et al., 2007*) but contrasts, at first glance, with the proportional increase of respiration to limb velocity observed in human participants (*Bechbache and Duffin, 1977*; *DiMarco et al., 1983*; *Casey et al., 1987*) (but see *Kay et al., 1975* for contradictory findings). It also raises the question of how mice cope with the increased metabolic demand associated with faster running and inclined route (*Heglund and Taylor, 1988*). In fact, it is likely that increased ventilation during running exercise may also operate through increased tidal volume (*Leith, 1976*). However, while our noticing of increased amplitude of diaphragm EMG signals during exercise denotes increased inspiratory muscular efforts and is compatible with increased tidal volume, our method cannot monitor further augmentations of respiratory airflows if supported by additional muscles. Accessory respiratory muscles in particular are active during high ventilatory demand such as exercise to expand and stabilize the chest wall (*Sieck and Gransee, 2012*). We therefore cannot exclude that, even at a stable respiratory frequency, additional muscles can be mobilized as displacement speed and effort intensity increase, to adjust ventilatory volumes on the fly and thereby help maintain respiratory homeostasis. Further experiments utilizing targeted recordings of accessory muscles (*Romer et al., 2017*) will be needed to evaluate this directly. Our data nonetheless demonstrate that any putative adjustments in volume not accessed here do not constrain the duration of the respiratory cycle, which remains constant over a

large range of displacement speeds. In the same line, the switch from passive to active expiration, a candidate signature of exercise hyperpnoea, mobilizes abdominal expiratory muscles (*Abdala et al., 2009*). This was only reflected in our data by manifest shortenings of expirations (Te), especially at gallop. Targeted recordings of expiratory abdominal muscles will be needed to ascertain it.

Interestingly, we found that respiratory rates increased further when animals engaged into the fastest, escape-like, running regimes in a linear corridor (*Figure 4*). Admittedly, our results cannot discriminate whether such higher respiratory rates compared to the treadmill situation reflect the higher displacement speeds or owe to the different protocol employed (air puff stimulus). This nevertheless indicates that the respiratory rhythm generator was not operating at maximal frequency regime during treadmill trotting but rather at a fixed frequency set point. Importantly, the strongest increase in respiratory frequency was seen when animals engaged into gallop, instead of trot, following the air puff. We therefore propose that efficient breathing adaptation to exercise may include, at least in mice, distinct dedicated behavior-dependent solutions. The transition from rest to moderate exercise could engage a default 'exercise' breathing at a fixed frequency set point compatible with a range of energetic demands corresponding to ritualistic territorial and explorative tasks and their share of uncertain evolutions (change of speed, of slope, etc…). There, the mobilization of other respiratory muscles may support on-the-fly adjustments of respiratory volumes. The engagement into faster displacement speeds, typically associated with escape behaviors, appears to operate through a further increase of respiratory frequency and, possibly, active expiration.

The mechanisms by which the breathing frequency is set, and those allowing transiting from it will need to be investigated. A bulk of evidence indicates that transitions from walk, to trot and gallop are controlled by specific circuits in the brainstem centers for locomotor initiation (*Bachmann et al., 2013*; *Caggiano et al., 2018*; *Josset et al., 2018*) and in the executive centers of the spinal cord (*Andersson et al., 2012*; *Talpalar et al., 2013*; *Bellardita and Kiehn, 2015*; *Skarlatou et al., 2020*). Therefore, it is likely that a direct modulation of the brainstem respiratory generator by a neuronal drive of central origin, that is from the brainstem and/or spinal locomotor centers, may trigger respiratory frequency increase during exercise (*Eldridge et al., 1981*; *Eldridge et al., 1985*; *Gariépy et al., 2012*; *Le Gal et al., 2014*; *Paterson, 2014*). Immediate concerns would be to identify candidate trigger neurons, as well as their partners in the respiratory centers. This should be achievable using selective interventional manipulations of locomotor and respiratory neuronal types, now both well-defined and accessible with molecular and genetic markers, viral tracers, and dedicated transgenic mice lines (*Bouvier et al., 2010*; *Talpalar et al., 2013*; *Ruffault et al., 2015*; *Kiehn, 2016*; *Del Negro et al., 2018*; *Skarlatou et al., 2020*).

# Materials and methods

### Key resources table

| Reagent type (species) or resource | Designation | Source or reference | Identifiers | Additional information |
|---|---|---|---|---|
| Strain strain background *Mus musculus* | C57BL/6J | Janvier Labs | IMSR Cat# JAX:000664, RRID:IMSR_JAX:000664 | https://www.janvier-labs.com/en/fiche_produit/c57bl-6jrj_mouse/ |
| Software, algorithm | Labscribe NI | iWorxs | | https://www.iworx.com/products/biomedical-engineering/labscribeni/ |
| Software, algorithm | Clampfit (pCLAMP 11) | Molecular Devices | RRID:SCR_011323 | https://www.moleculardevices.com/products/axon-patch-clamp-system/acquisition-and-analysis-software/pclamp-software-suite#Resources |
| Software, algorithm | Deeplabcut | Mathis Lab | | http://www.mousemotorlab.org/deeplabcut |
| Software, algorithm | ToxTrac | *Rodriguez et al., 2017* | | https://sourceforge.net/projects/toxtrac/ |
| Software, algorithm | GraphPad Prism 7 | | RRID:SCR_002798 | https://www.graphpad.com/ |
| Software, algorithm | 2nd Look | IO Industries | | https://shop.ioindustries.com/products/2ndlook |

## Animals

All experiments were conducted in accordance with EU directive 2010/63/EU and approved by the local ethical committee (authorization 2020–022410231878). Experiments were performed on C57BL/6J mice of either sex, aged 3 months at the time of the EMG implantation and obtained from Janvier Labs (Le Genest-Saint-Isle, France). All animals were group-housed in plastic breeding cages with free access to food and water, in controlled temperature conditions and exposed to a conventional 12 hr light/dark cycle. Animals were managed by qualified personnel and efforts were made to avoid suffering and minimize the number of animals.

## Housing and training protocol

Mice were divided in two groups: untrained mice housed in normal environments, and trained mice that had free access to a running-wheel in the cage. Furthermore, trained mice were exercised daily, from the age of 1 month, for 8 consecutive weeks (see *Figure 5A*) on a custom-made motorized treadmill with adjustable speed range (Scop Pro, France, belt dimensions: 6 cm x 30 cm). Each training session consisted in placing the animals on the stationary treadmill with no incline for 5 min, before they were engaged for a total of 15 min of running at three distinct speeds (25, 40, and 50 cm/s, 5 min at each speed). Mice could rest for 5 min after running before being placed back in their cage. A week after the EMG implantation, both groups were exercised daily for a week using a similar paradigm but for 1 min at each speed. This step was crucial to obtain stable running animals during experimental sessions. Only animals determined to be good runners based on their ability to run without reluctance were included in this study.

## Diaphragm EMG recordings

### Fabrication of EMG electrodes

The protocol was inspired by previous work (*Pearson et al., 2005*). The electrodes were made of Teflon-coated insulated steel wires with an outside diameter of 0.14 mm (A-M systems, ref 793200). For each animal, a 12 cm pair of electrodes was prepared as follows (*Figure 1A*). Two wires were lightly twisted together, and a knot was placed 5 cm from one end. At 1 cm from the knot, the Teflon insulation was stripped over 1 mm from each wire so that the two bared regions were separated by about 2 mm. The ends of the two wires were soldered to a miniature dissecting pin. The free ends of the electrodes, as well as a 5 cm ground wire, were soldered to a micro connector (Antelec). Nail polish and an insulation sleeve were used to insulate the wires at the connector.

### Surgical implantation of EMG electrodes in the diaphragm

To implant the diaphragm, 3-month-old animals were anaesthetized using isoflurane (4% at 1 L/min for induction and 2% at 0.2 L/min for maintenance), placed in a stereotaxic frame (Kopf) and hydrated by a subcutaneous injection of saline solution (0.9%). Their temperature was maintained at 36°C with a feedback-controlled heating pad. This step was crucial to ensure post-surgery survival. The skull was exposed and processed to secure the micro connector using dental cement (Tetric Evofow). The ground wire was inserted under the neck's skin and the twisted electrodes were tunneled towards the right part of the animal guided by a 10-cm silicon tube of 2 mm inner diameter. The animal was then placed in supine position, the peritoneum was opened horizontally under the sternum, extending laterally to the ribs, and the silicon tube containing the electrodes was pulled through the opening. The sternum was clamped and lifted upwards to expose the diaphragm. A piece of stretched sterile parafilm was placed on the upper part of the liver to avoid friction during movement of the animal and to prevent conjunctive tissue formation at the recording sites. The miniature dissecting pin was pushed through the right floating ribs. The pin was then inserted through the sternum, leaving the bare part of the wires in superficial contact with the diaphragm (*Figure 1B*). The position of the electrodes was secured on both sides of the floating ribs and sternum using dental cement. The pin was removed by cutting above the secured wires. The peritoneum and abdominal openings were sutured and a head bar was placed on the cemented skull to facilitate animal's handling when connecting and disconnecting EMG cables during behavioral sessions. Buprenorphine (0.025 mg/kg) was administered subcutaneously for analgesia at the end of the surgery and animals were observed daily following the surgery.

## Behavioral experiments

Upon a full week of recovery, implanted animals were connected with custom light-weight cables to an AC amplifier (BMA-400, CWE Inc) and neurograms were filtered (high-pass: 100 Hz, low-pass: 10 KHz), collected at 10 kHz using a National Instruments Acquisition card (USB-6211) and live-integrated using the LabScribe NI software (iWorxs).

### Plethysmography recordings

Four unrestrained implanted animals were connected to the AC amplifier and placed inside a hermetic whole-body plethysmography chamber (*Ruffault et al., 2015*), customized to allow the passage of the EMG cable. Respiratory volume and diaphragm EMG activity were recorded simultaneously over a period of 10 min using the LabScribe NI software. Respiratory volume was derived to obtain the respiratory flow (*Figure 1C,D*).

### Open field experiments

For analyzing changes in global locomotor activity, control or implanted and connected animals were placed, one at a time, in a custom open field arena (opaque PVC, 60 × 70 cm) without prior habituation and filmed from above during 10 min at 20 frames/s using a CMOS camera (Jai GO-5000-C-USB). The open-source software ToxTrac (*Rodriguez et al., 2017*) was used for automated tracking of the animal's position over time. Following geometric calibration, the following parameters were extracted for each mice (*Figure 1H*): the mobility rate defined as the percent of time that the animal spent moving above 7.5 cm/s, the average mobility speed defined as the average speed of the animal during mobility (i.e. above 7.5 cm/s) and the total distance travelled per 10 min recording. Within each category (i.e. control or EMG-implanted) a grand-mean ± SD across mice was then calculated to produce histograms.

### Running experiments

We used a custom-built treadmill (Scop Pro, France) whose speed was remotely adjusted using a USB servo controller (Maestro Polulu). Implanted untrained (n = 7) and trained (n = 3) animals followed the same running experiments while recording breathing changes. First, we evaluated breathing changes during running at trot, without or with incline. For each speed, the protocol was as follows. Animals were first placed on the stationary treadmill to monitor basal respiration. Animals were then challenged to trot at the lowest speed (15 cm/s) for 1.5 min followed by a 5-min break. The treadmill was then inclined with a 10% slope and animals exercised at the same speed for 1.5 min. This sequence was repeated for the three other speeds (25, 40, and 50 cm/s) with 5 min of rest between trials. At the end of the running experiment, a breathing recovery period was recorded. We also recorded breathing changes throughout a continuous 10 min exercise at 40 cm/s on the motorized treadmill for n = 2 untrained and n = 2 trained animals. Since faster running speeds typically associated with escape cannot be achieved on the treadmill without aversive conditioning (*Fernando et al., 1993*), we resorted to air puff induced escape in a linear corridor (80 × 10 cm) as previously described (*Talpalar et al., 2013*; *Caggiano et al., 2018*). Trials were classified as trot or gallop based on the alternation (trot) or synchronization (gallop) of the hindlimbs. The test was repeated, with several minutes of rest between trials, until enough bouts (around 20) were acquired.

During running sessions, animals were filmed from the side at 200 fps and 0.5 ms exposure time using a CMOS camera (Jai GO-2400-USB) and images were streamed to a hard disk using the 2nd LOOK software (IO Industries). The start of the EMG recordings was hardware-triggered by the start of the video-recordings using the frame exposure output of the video camera, so that the two recordings are synchronized.

### Automated analysis of limb movements

To track limb movements during running, we used DeepLabCut (version 2.1.5.2, *Mathis et al., 2018*, see *Figures 3,4*). We manually labeled the positions of the head and the 4 paws from 50 frames of each video. We then used 95% of the labeled frames to train the network using a ResNet-50-based neural network with default parameters for three training iterations. We validated with two shuffles and found that the test error for trot experiments was: 3.33 pixels and the train error: 2.37 pixels (image size: 1344 × 301). Similarly, we trained the network for gallop condition using a

ResNet-50-based neural network with default parameters for one training iteration. We validated with two shuffles and found that the test error was: 3.43 pixels and the train error: 2.43 pixels (image size: 1936 × 230). These networks were then used to analyze videos from similar experimental settings. X and Y coordinates from the head and the four limbs were then extracted and interpolated to 10 kHz to match the EMG recordings. The latter were exported from LabScribe to Clampfit (Molecular Devices), and both sets of signals were merged in a single file, before being processed offline in Clampfit.

## Quantifications and statistics

The instantaneous frequency and amplitude of respiratory bursts were detected using the threshold search in Clampfit from stable trotting moments, that is when the animal's speed was in phase with the treadmill, inferred by the absence of changes in head's horizontal coordinates. Instantaneous respiratory frequency was measured for a total duration of 6 s, thus encompassing around 60 cycles. These measurements were either done using two to three windows taken at any stable moment of the 1.5-min run (excluding the first 20 s to exclude possible stress-induced changes when the treadmill is just engaged), and averaged to give the value for each animal. For the analysis of longer runs (*Figure 2—figure supplement 1*), similar measurements were done at 5 and 10 min of continuous running. Inspiratory time (Ti) was defined as the duration of the diaphragmatic burst and the expiratory time (Te) as the silent period in between bursts as illustrated in *Figure 1D*. Respiratory amplitude change was normalized and expressed as a percent of the control amplitude before running started. Values for respiratory bursts (frequency, Ti, Te and amplitude) were expressed as mean ± SD across n animals. Statistical differences between means were analyzed using Mann-Whitney *U* tests (GraphPad Prism 9) and changes were considered as not significant (ns) when $p > 0.05$ and as significant when $p < 0.05$. Significance was reported as * when $0.05 < p < 0.01$, ** when $0.01 < p < 0.001$ and as *** when $p < 0.001$. All p values are declared in figures.

The temporal coordination of breaths to strides was represented with circular statistics, imprinted from numerous studies having investigated the cycle-to-cycle correlations of motor activities (*Kjaerulff and Kiehn, 1996*; *Talpalar et al., 2013*; *Skarlatou et al., 2020*). The phase of each individual inspiratory burst within the locomotor cycle (ΦInsp, at least 35 bursts on the treadmill and at least 20 for the air puff context) is represented as the position, from 0 to 1, of the black diamond marks on the outer circle (see *Figure 3C*; *Figure 3D*). For each animal, we also computed the mean phase of consecutive inspiratory bursts and represented it as a colored dot (the mean phases of different animals are in different colors). The distance R of the mean phase to the center of the circle indicates the concentration of individual phase values around the mean, as established by *Kjaerulff and Kiehn, 1996*. If inspiratory and locomotor movements are temporally correlated, then individual phase values will be concentrated around a preferred phase value (for instance 0 or 1, at the top of the circle, if the two motor activities were temporally locked). The mean value would then be positioned at a significant distance from the center. Conversely, if inspiratory and locomotor movements are not coupled, individual phases will be evenly distributed across the circle. Consequently, the mean phase value will be at a short distance from the diagram center, illustrating the dispersion of values around the mean. The inner circles of the circular diagrams depict the threshold for mean phase values to be considered significantly oriented (R < 0.3) as commonly done (*Kjaerulff and Kiehn, 1996*; *Talpalar et al., 2013*; *Skarlatou et al., 2020*). Circular plots were obtained using a custom macro in Excel. Auto and cross-correlograms (*Figure 3—figure supplement 2*) were computed in Clampfit for one representative animal during stable epochs of running, inferred by the absence of changes in head's horizontal coordinates.

## Acknowledgements

This work was funded by an Agence Nationale de la Recherche grant to JB (ANR-17-CE16-0027) and supported by CNRS, Université Paris-Saclay and NeuroPSI. CH holds doctoral fellowships from Région Ile-de-France and Fondation pour la Recherche Médicale. We thank the animal facility for housing animals, Edwin Gatier and Anthony Renard for help with DeepLabCut and Aurélie Heuzé for lab management and help with animals.

## Additional information

### Funding

| Funder | Grant reference number | Author |
|---|---|---|
| Agence Nationale de la Recherche | ANR-17-CE16-0027 | Julien Bouvier |
| Fondation pour la Recherche Médicale | Doctoral fellowship | Coralie Hérent |
| Région Ile-de-France | Doctoral fellowship | Coralie Hérent |
| University of Paris-Saclay | | Julien Bouvier |
| Centre National de la Recherche Scientifique | | Julien Bouvier |
| NeuroPSI | | Julien Bouvier |

The funders had no role in study design, data collection and interpretation, or the decision to submit the work for publication.

### Author contributions

Coralie Hérent, Conceptualization, Formal analysis, Investigation, Methodology, Writing - review and editing; Séverine Diem, Formal analysis, Investigation, Methodology; Gilles Fortin, Conceptualization, Writing - review and editing; Julien Bouvier, Conceptualization, Formal analysis, Supervision, Funding acquisition, Methodology, Writing - original draft, Project administration, Writing - review and editing

### Author ORCIDs

Coralie Hérent (ID) https://orcid.org/0000-0002-8472-5097
Gilles Fortin (ID) https://orcid.org/0000-0002-2123-8603
Julien Bouvier (ID) https://orcid.org/0000-0002-1307-4426

### Ethics

Animal experimentation: All experiments were conducted in accordance with EU directive 2010/63/EU and approved by the local ethical committee (authorization 2020-022410231878). Animals were managed by qualified personnel and every effort was made to minimize suffering and reduce the number of animals.

### Decision letter and Author response

Decision letter https://doi.org/10.7554/eLife.61919.sa1
Author response https://doi.org/10.7554/eLife.61919.sa2

## Additional files

### Supplementary files

• Transparent reporting form

### Data availability

All data generated or analysed during this study are included in the manuscript and supporting files. Source data files have been provided for all figures.

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
