## [Decision Letter]

**Acceptance summary:**

This study examines the timing of breathing and limb movements in mice during running to determine if there is an obligatory coordination, which is generally thought to be important for the respiratory adaptations during running in various species. The authors' analyses show that breathing frequency increases as required but without synchronization to limb movements at any running speed or locomotor gait. These results provide a remarkable counterexample to the concept that breathing and locomotor rhythms become naturally coupled in a relatively fixed manner during running, and introduce the mouse model as part of the toolbox to investigate how breathing is augmented during running exercise.

**Decision letter after peer review:**

Thank you for submitting your article "Independent respiratory and locomotor rhythms in running mice" for consideration by *eLife*. Your article has been reviewed by three peer reviewers, including Jeffrey C Smith as the Reviewing Editor and Reviewer #1, and the evaluation has been overseen by Ronald Calabrese as the Senior Editor. The following individual involved in review of your submission has agreed to reveal their identity: William K Milsom (Reviewer #3).

The reviewers have discussed the reviews with one another and the Reviewing Editor has drafted this decision to help you prepare a revised submission.

Summary:

In their Short Report, the authors examine the interactions between locomotor and respiratory rhythms in adult mice. These are technically well executed experiments with novel results about the patterning of inspiratory activity in relation to locomotor speed and gait important for understanding the dynamics of exercise hyperpnea in mice. The data analyses and presentation are excellent. The experimental results show that the respiratory rate increases during the trot, but independently of the locomotor speed of the limbs and the workload (i.e., treadmill with incline). They also report the absence of locomotor-respiratory rhythm coupling, even when the animals run faster at a gallop (i.e., when the gait changes), or when animals have been trained, thus accounting for experience level. These results in mice potentially provide an interesting counterexample to what previous studies have described about the synchronization of breathing and locomotor rhythms during running in other quadruped mammals, but there are important caveats. There are some concerns about experimental design, including that the experiments were carried out on a limited range of locomotor speeds, and about the context in which the data are presented with regard to existing literature, all of which requires additional interpretations and discussion of the results. The authors should address all of the concerns listed below in a revised manuscript with an expanded Discussion.

1) In the Abstract and elsewhere in the manuscript, the authors imply that they are addressing the general question of whether breathing frequency is inherently proportional to limb velocity and/or imposed by a synchronization of breaths to strides. Since it is well established that many quadruped mammals can synchronize respiratory and locomotor cycles over a wide range of running speeds (which the authors acknowledge), the outcome can be dependent on the species examined. The authors need to emphasize more in the Discussion that while their conclusion about obligatory involvement of hardwired neural mechanisms may be accurate in the present experiments, how the respiratory adaptation to exercise (locomotion) is implemented in mice does not necessarily represent a general principle.

2) Related to point 1, the authors' observation that respiratory rate increases during running to a fixed and stable value, irrespective of trotting velocities and of inclination may reflect species related mechanical constraints. To what extent is the 'fixed' increase in respiratory frequency reflective of this animal's pulmonary mechanics? You do show that the amplitude of your EMGs increases with exercise and point out that these are not a good indicator of tidal volume – but this is an animal in which increases in tidal volume may be the most economical way of increasing ventilation. The mouse's chest wall is almost infinitely compliant, and recoils inward over most of the range of lung volumes. Functional residual capacity and residual volume are the same. These animals lack an expiratory reserve volume and with an open airway, the chest wall collapses inward and rests on the lung. (Leith, 1976). It is quite conceivable that taking larger volumes puts constraints on the time needed for inspiration and expiration which may influence the timing of locomotor stride and breathing frequencies? Some discussion along these lines would be useful.

3) With mice there can be significant strain differences. You give a rationale for choosing mice for this study based on factors other than exercise capacity. Would you consider mice good runners? Would you consider respiratory-locomotor entrainment to be more likely in sprinters or endurance runners? In light of this (that endurance, sustained running may be more likely to promote respiratory-locomotor entrainment), having chosen mice, you should discuss strain differences in exercise capacity and possible implications for respiratory-locomotor entrainment. Strain differences have been shown to give rise to two-fold differences in time to exhaustion and almost 10-fold differences in work performed (Avila, Kim and Massett, 2017).

4) Even within the same strain, some mice show high voluntary wheel running compared to others. High runners run approximately three times as many revolutions per day. Not all mice are equal. (Meek et al., 2009). You only ran your animals for 15-minute trials. This and point 2 above may help explain why entrainment is not seen in your mice. Is it possible that you are looking for it in a couch potato when it might be present in the Jacques Anquetil of the mouse world?

5) The authors state as a main conclusion that during running (in your mouse strain), respiratory rate is independent of locomotor speeds (also mentioned in article title). This statement is a bit confusing since the authors report a clear significant difference in the respiratory rate between the trot and the gallop (Figure 4C), the latter being assumed to be generated at faster speed than the former. This needs to be clarified.

6) During a given gait (i.e., trot), the authors conclude that the observed increase in respiratory rate is independent of locomotor speed (25, 40 and 50 cm/s) and intensity of effort (i.e., treadmill with incline). However, the ability of this strain of mice to produce trot from 5 to 75 cm/s (and even more) on a treadmill has been clearly reported in a previous detailed locomotor study (see Lemieux et al., 2016). Therefore, the possibility that the animals would generate different respiratory rates that might show synchronization at slower or faster trotting speeds cannot be excluded. The conclusion about the lack of synchronization needs to be qualified, otherwise analysis over a wider range of trotting speeds is necessary.

7) It is difficult to understand the reasons why different protocols have been used to trigger trot (treadmill) and gallop (air puff). With air puff, episodes of bound (rather than gallop; see again Lemieux et al., 2016) are of short duration, which may be too short for comparing to data obtained during trotting on treadmill. Moreover, the possibility that respiratory rate would be different during air puff-induced bound and treadmill-induced bound cannot be excluded. Also, using treadmill to induce hopping-like running would allow to explore a wider range of 'gallop' speeds. Furthermore, the authors state that breaths are not temporally coordinated to strides in mice (during trot and bound). But maybe locomotor-respiratory coupling would occur during faster trot or during a longer bout of hopping-like running at different speeds (from 75 to 150 cm/s, see Lemieux et al.). Here again, experiments conducted over a wider range of trotting and 'hopping-like' regimes on a treadmill (and not with air puff) would need to be performed before such conclusions can be drawn. These issues should be discussed.

8) The reviewers agree with you that with the mice you used, and the protocols you used, the mice arguably show little signs of entrainment. However, entrainment is not always 1:1 and indeed not always even integers (2:1, 3:1 etc.). There are reports of 5:2 or 5:3 ratios. These ratios are always given as locomotor events : respiratory events. Examining Figures 3 and 4 of this manuscript one could argue that there is a 3:5 ratio apparent in both the trot and the gallop – 5 breaths for every three strides. The literature is out there and should be referenced. (for instance: Persego and Viala 1991. J Physiol (Paris) 85:38-43). Have the authors analyzed higher order coupling ratios?

9) From a physiological point of view, the absence of significant differences in respiratory parameter values (FR, TI, TE, amplitude of respiratory discharges) during different trotting speeds (on a treadmill with or without incline) is unexpected and surprising. Indeed, when locomotor speed accelerates and/or when the intensity of effort increases (treadmill with incline), it is expected that the metabolic demand is also increased. How does the mouse satisfy the increasing energetic demand if neither the respiratory frequency nor the tidal volume are changed when the locomotor speed is augmented? This major point, however, is not discussed by the authors and should be addressed.

Revisions expected in follow-up work:

In subsequent studies it would be important for the authors to further validate their conclusions that breaths are not temporally coordinated to strides in mice (during trot, gallop, and other gaits such as bound) by performing studies with a single experimental paradigm (i.e., treadmill running) to produce different locomotor gaits over a wider range of speeds and durations of locomotor regimes. This would allow further testing if locomotor-respiratory coupling occurs during faster trot or during a longer bout of gallop (and other gaits) running at higher speeds (i.e., above 50 cm/s, see Lemieux et al.) than tested in the current studies.

---

## [Author Response]

Revisions for this paper:1) In the Abstract and elsewhere in the manuscript, the authors imply that they are addressing the general question of whether breathing frequency is inherently proportional to limb velocity and/or imposed by a synchronization of breaths to strides. Since it is well established that many quadruped mammals can synchronize respiratory and locomotor cycles over a wide range of running speeds (which the authors acknowledge), the outcome can be dependent on the species examined. The authors need to emphasize more in the Discussion that while their conclusion about obligatory involvement of hardwired neural mechanisms may be accurate in the present experiments, how the respiratory adaptation to exercise (locomotion) is implemented in mice does not necessarily represent a general principle.

We acknowledge that we should have been clearer about what is known, or not, in other quadrupeds and about the fact that what we report in mice may not be valid for all quadrupeds. However, we somewhat disagree that “it is well established that many quadruped mammals can synchronize respiratory and locomotor cycles over a wide range of running speeds”. We now highlight, in an expanded Discussion, the ambiguities in previous work on that matter.

Specifically, while the reviewer is correct that the existence of a synchronization is stated in multiple reviews or research articles, the interpretation of “synchronizing” is very confusing across the literature. We now make clear in the discussion that the indicators often used (the “Locomotor-respiratory coupling LRC”, or a so-called “respiratory entrainment”) do not necessarily imply that breaths and strides are synchronized on a cycle-to-cycle basis:

“It must be stressed however that the indicator that is most typically used, the “locomotor respiratory coupling” (or LRC), has a very ambiguous interpretation. In many instances, it only reports the frequency ratio between the two movements, i.e. locomotor events per respiratory events (Bechbache and Duffin, 1977; Bramble and Carrier, 1983; Paterson et al., 1986; Corio et al., 1993). […] For a temporal locking of breaths and strides, a synchronizing neural command needs to be produced from either active respiratory or the locomotor networks and be fed onto respectively the locomotor or respiratory networks.”

This altogether leaves open the possibility of generalization of our findings in mice. This is actually a strong message of our work that we hope we made clearer.

2) Related to point 1, the authors' observation that respiratory rate increases during running to a fixed and stable value, irrespective of trotting velocities and of inclination may reflect species related mechanical constraints. To what extent is the 'fixed' increase in respiratory frequency reflective of this animal's pulmonary mechanics? You do show that the amplitude of your EMGs increases with exercise and point out that these are not a good indicator of tidal volume – but this is an animal in which increases in tidal volume may be the most economical way of increasing ventilation. The mouse's chest wall is almost infinitely compliant, and recoils inward over most of the range of lung volumes. Functional residual capacity and residual volume are the same. These animals lack an expiratory reserve volume and with an open airway, the chest wall collapses inward and rests on the lung. (Leith, 1976). It is quite conceivable that taking larger volumes puts constraints on the time needed for inspiration and expiration which may influence the timing of locomotor stride and breathing frequencies? Some discussion along these lines would be useful.

We agree that increased ventilation may likely be operated through increased Vt. However, we do not have a direct measurement of ventilatory volumes with EMG diaphragmatic recordings. We had already touched upon this limitation but we now unpack on this further in the Discussion:

“[this increased breathing rate to a stable value] also raises the question of how mice cope with the increased metabolic demand associated with faster running and inclined route (Heglund and Taylor, 1988). […] Our data nonetheless demonstrate any putative adjustments in volume not accessed here do not constrain the duration of the respiratory cycle that remains constant over a large range of displacement speeds”.

3) With mice there can be significant strain differences. You give a rationale for choosing mice for this study based on factors other than exercise capacity. Would you consider mice good runners? Would you consider respiratory-locomotor entrainment to be more likely in sprinters or endurance runners? In light of this (that endurance, sustained running may be more likely to promote respiratory-locomotor entrainment), having chosen mice, you should discuss strain differences in exercise capacity and possible implications for respiratory-locomotor entrainment. Strain differences have been shown to give rise to two-fold differences in time to exhaustion and almost 10-fold differences in work performed (Avila, Kim and Massett, 2017).

In line with this comment, and the related concerns that what applies in mice, and all the more in a single mouse strain, may not be a general principle – which we clearly do not dispute – we have substantially revised the Discussion and notably included the following reasoning:

“Admittedly, the complete absence of synchronization of breaths to strides we report here in selected contexts in the C56BL/6 mouse strain does not ascertain that synchronization never occurs in mice. […] This possibility is also supported by the persistence of a normal ventilatory response to exercise following the removal of peripheral signals in animal models (Eldridge et al., 1981; Eldridge et al., 1985) and human subjects (Fernandes et al., 1990).”

4) Even within the same strain, some mice show high voluntary wheel running compared to others. High runners run approximately three times as many revolutions per day. Not all mice are equal. (Meek et al., 2009). You only ran your animals for 15-minute trials. This and point 2 above may help explain why entrainment is not seen in your mice. Is it possible that you are looking for it in a couch potato when it might be present in the Jacques Anquetil of the mouse world?

This comment relates to point (3) above and the quoted Discussion paragraph applies here as well. We acknowledge that, similar to the situation in human subjects, inter-animal variability may occur in mice. We should stress however that animals included in the analysis were considered to be good runners on the basis of their ability to run without reluctance, as now stressed in the Materials and methods.

Secondly, we admit that our training protocol is moderate. However, giving free access to a running wheel in the cage (without treadmill training) for 3 to 4 weeks (which is shorter than the 8 weeks we do here) was shown to have multiple benefits typically associated with exercise, including: enhanced locomotor skills and changes in neurotransmitter release (Li and Spitzer, 2020), anti-inflammatory actions leading to improved cognitive function in mouse models of AD (Parachikova et al., 2008), and to alter sleep cycles (Lancel et al., 2003). We have combined it with a daily treadmill running for 15 min at 3 different speeds. Therefore this training; although not as demanding as the protocol of Avila et al., who trained the mice until exhaustion, does affect the animal’s physiology and locomotor skills. We provide further details on our training method in the Results and acknowledge the limitation in the Discussion.

5) The authors state as a main conclusion that during running (in your mouse strain), respiratory rate is independent of locomotor speeds (also mentioned in article title). This statement is a bit confusing since the authors report a clear significant difference in the respiratory rate between the trot and the gallop (Figure 4C), the latter being assumed to be generated at faster speed than the former. This needs to be clarified.

It is true that at gallop, the respiratory rate is higher than on the treadmill. In the revised manuscript, we now also show that breathing rate is slightly higher during air-puff induced trot (~70 cm/s), as compared to the maximum speed we could reach on the treadmill (50 cm/s). This could indicate that respiratory frequency is only independent on locomotor speed at moderate regimes. We have therefore toned down the conclusion that “respiratory rate is independent of running speed” throughout the paper.

In our original title, we used the word rhythm (to refer to when events occur in time, i.e., the timing of events), not speed (how many events occur per unit of time). By “independent rhythms” we thus meant that the timing of breaths was not dependent on the timing of strides. We understand the confusion though since in the field, rhythm is commonly used as the correlate of speed (while the term pattern often relates to the temporal structure of events). To make it more clear here that our work demonstrates that changes in speed (from rest to running, or from trotting to gallop) take place without a temporal phasing of breaths to strides, we have changed the title to: “Absent phasing of respiratory and locomotor rhythms in running mice”.

6) During a given gait (i.e., trot), the authors conclude that the observed increase in respiratory rate is independent of locomotor speed (25, 40 and 50 cm/s) and intensity of effort (i.e., treadmill with incline). However, the ability of this strain of mice to produce trot from 5 to 75 cm/s (and even more) on a treadmill has been clearly reported in a previous detailed locomotor study (see Lemieux et al., 2016). Therefore, the possibility that the animals would generate different respiratory rates that might show synchronization at slower or faster trotting speeds cannot be excluded. The conclusion about the lack of synchronization needs to be qualified, otherwise analysis over a wider range of trotting speeds is necessary.

We acknowledge that this point deserves clarification and we have added new analysis and new figure panels on that matter. In our hands, and in line with previous reports (e.g., Fernando, Bonen and Hoffman-Goetz, 1993), it was virtually impossible to challenge mice to run at higher speeds than 50 cm/s on a treadmill without prior training with an electrified grid at the rear. We specifically aimed at avoiding any form of training, especially with aversive conditioning or motivation, so that we could document the “naive”, volitional, running situation (presumably reflecting a hardwired circuit), and eventually compare it to that after training.

For the revised manuscript, we have thus analyzed respiratory frequencies, and the synchronization with strides, when animals engage in trotting following the air-puff. This is now illustrated in the new main Figure 4A-C and G-J, and covered in the revised Results sections retitled “Breathes are not temporally correlated to strides at higher displacement speeds, including at gallop, in a linear corridor”.

In this paradigm, animals trot at a higher speed (72 cm/s) than the maximum we could achieve on the treadmill, and their respiratory frequency is slightly higher than at trot on the treadmill (Figure 4G). Our results do not speak to whether this higher breathing rate owes to the protocol used to engage running that differs from the treadmill, or to the faster displacement speed. With this limitation in mind, and the other comments raised on this issue, we have thus toned down our conclusion that breathing frequency is independent of trotting velocities. We maintain that it is independent between 15 to 50 cm/s, on the treadmill, but cannot exclude that escape behaviors, whether achieved at trot of gallop, may require faster breathing rate. Importantly however, there was still no synchronization of breaths to strides at this faster trotting induced by air-puff (Figure 4B, C). We think that this latter addition highlights that the lack of such synchronization is likely not dependent on the running speed, protocol, or environment.

Furthermore, we have also added a lower running speed on the treadmill: 15 cm/s (Figure 2). This speed was included in our testing protocol but we had initially excluded it from the paper, since, at this pace, animals often mobilize various motor gaits (exemplified in the Lemieux et al. study) and their running is often not stable. Yet for the revised version, we have included analysis of respiratory frequencies and synchronization at 15 cm/s for 4 animals, both on the flat and inclined treadmill (the 3 others were too unstable at 15 cm/s). There, respiratory frequency was similar to that at the higher trotting speeds on the treadmill. There was also no synchronization of breaths to strides. This new data is now presented in the Results and in dedicated panels in the main figures (frequency: Figure 2B-E, G-J; synchronization: Figure 3D).

We have not analyzed running at lower speeds than 15 cm/s. There, the mice only show short walking bouts interleaved with episodes of grooming or exploration, with sniffing and a lot of whisker sweeps on the side walls (e.g., Arkley et al., 2014, Strategy change in vibrissal active sensing during rat locomotion. Curr Biol 24, 1507-1512.). These latter behaviors, particularly whisking, also mobilize the respiratory apparatus, altogether precluding a stable analysis of respiratory frequency linked with exercise. We have clarified this in the Results section. We also discuss limitations associated with the examined range of velocities further in the expanded Discussion.

7) It is difficult to understand the reasons why different protocols have been used to trigger trot (treadmill) and gallop (air puff). With air puff, episodes of bound (rather than gallop; see again Lemieux et al., 2016) are of short duration, which may be too short for comparing to data obtained during trotting on treadmill. Moreover, the possibility that respiratory rate would be different during air puff-induced bound and treadmill-induced bound cannot be excluded. Also, using treadmill to induce hopping-like running would allow to explore a wider range of 'gallop' speeds. Furthermore, the authors state that breaths are not temporally coordinated to strides in mice (during trot and bound). But maybe locomotor-respiratory coupling would occur during faster trot or during a longer bout of hopping-like running at different speeds (from 75 to 150 cm/s, see Lemieux et al.). Here again, experiments conducted over a wider range of trotting and 'hopping-like' regimes on a treadmill (and not with air puff) would need to be performed before such conclusions can be drawn. These issues should be discussed.

A lot of answers to this point are given to the above comment. As explained, we could not engage mice into faster running, and never into gallop, on the motorized treadmill. We believe that specific training, notably using aversive motivation at the rear, is required, together with a longer treadmill belt to allow gallop. We aimed at specifically avoiding training or aversive motivations, so that we could document the “naive” situation (presumably reflecting a hardwired circuit), and eventually compare it to that after training. This is the reason why we have used the linear corridor with the air-puff, which has been used before (e.g. Talpalar et al., 2013; Caggiano et al., 2018). In the revised version, we have also analyzed breaths/strides coordination at trot, using the air-puff, which engages faster displacement speeds than what we could achieve on the treadmill. Yet, similar to the treadmill, we found no breaths/strides synchronization. Therefore, while we can only agree that this is a different protocol than the treadmill, it does not seem to make a difference for the lack of synchronicity. We think that this is actually an asset, by highlighting that the lack of synchronization on the treadmill is likely not a treadmill-specific feature, but likely applies to other running contexts.

On top of new analyses, the Discussion has been substantially revised to incorporate these elements.

Of course, we also admit that the episodes of gallop are short when induced by air-puff. Unfortunately, we cannot have mice galloping on our treadmill. We thus cannot exclude that sustained galloping might favor the coupling. This is now also touched upon in the Discussion.

8) The reviewers agree with you that with the mice you used, and the protocols you used, the mice arguably show little signs of entrainment. However, entrainment is not always 1:1 and indeed not always even integers (2:1, 3:1 etc.). There are reports of 5:2 or 5:3 ratios. These ratios are always given as locomotor events : respiratory events. Examining Figures 3 and 4 of this manuscript one could argue that there is a 3:5 ratio apparent in both the trot and the gallop – 5 breaths for every three strides. The literature is out there and should be referenced. (for instance: Persego and Viala 1991. J Physiol (Paris) 85:38-43). Have the authors analyzed higher order coupling ratios?

We have substantially expanded our Discussion, as detailed also for the related comment (1). We had already included numerous citations of previous work having established, or not, a so-called “coupling” or “entrainment”, both in humans and animal models. We would like to specify here our point of view. As explained earlier, the terms “entrainment”, or “coupling” are being used very confusingly in the literature, and we intentionally did not use them in our study. As we now detail in the Discussion, and as raised by the reviewer here, these terms often reflect the locomotor events per respiratory events. In other words, a specific “coupling” 2:1 indicates, in most publications, two strides per breaths, on average. This is the ratio between respiratory and locomotor frequencies, but it says nothing about the synchronicity of both movements on a cycle-to-cycle basis, which is what we looked at here. Only some studies did investigate the cycle-to-cycle coordination as we did here, using the term “LRC degree”, or “% of LRC”, and these studies reveal that the coordination is far from being the rule. This is both true in humans (Kay et al., 1975; Bernasconi and Kohl, 1993; Daley et al., 2013; Stickford et al., 2015), horses (Lafortuna et al., 1996) and cats (Iscoe et al., 1981).

In our revised Discussion, copied as an answer to comment (1), we highlight in detail this ambiguity associated with the terms “LRC”, “coupling” and “entrainment”.

As for the possibility that breaths and strides might be synchronized regularly, but not at every cycle, we now provide cross-correlation plots between the limb and the diaphragm for one example animal. This is given as a figure supplement for Figure 3, for the 3 main speeds tested on the treadmill, both flat and inclined. All cross-correlograms are flat, indicating that there is no form of rhythmic common modulation between strides and breathes, excluding a phase coupling at higher ratios. Admittedly, we could not do these cross-correlograms in the air-puff situation since there are too few consecutive bouts.

9) From a physiological point of view, the absence of significant differences in respiratory parameter values (FR, TI, TE, amplitude of respiratory discharges) during different trotting speeds (on a treadmill with or without incline) is unexpected and surprising. Indeed, when locomotor speed accelerates and/or when the intensity of effort increases (treadmill with incline), it is expected that the metabolic demand is also increased. How does the mouse satisfy the increasing energetic demand if neither the respiratory frequency nor the tidal volume are changed when the locomotor speed is augmented? This major point, however, is not discussed by the authors and should be addressed.

Indeed, the energetic cost of locomotion is proportional to the stride frequency, and thus to displacement speed (Heglund and Taylor, 1988). While our results indicate little relation between breathing frequency and displacement speed, part of the respiratory adaptation may indeed operate through increased tidal volume (VT). As we acknowledge, our recording method does not provide a direct measurement of VT, even more so considering that changes in VT could be supported by other factors and muscles than only the amplitude of the diaphragmatic drive. We had already touched upon this limitation but we now unpacked on this further in the Discussion:

“It also raises the question of how mice cope with the increased metabolic demand associated with faster running and inclined route (Heglund and Taylor, 1988). […] Our data nonetheless demonstrate any putative adjustments in volume not accessed here do not constrain the duration of the respiratory cycle that remains constant over a large range of displacement speeds.”

Revisions expected in follow-up work:In subsequent studies it would be important for the authors to further validate their conclusions that breaths are not temporally coordinated to strides in mice (during trot, gallop, and other gaits such as bound) by performing studies with a single experimental paradigm (i.e., treadmill running) to produce different locomotor gaits over a wider range of speeds and durations of locomotor regimes. This would allow further testing if locomotor-respiratory coupling occurs during faster trot or during a longer bout of gallop (and other gaits) running at higher speeds (i.e., above 50 cm/s, see Lemieux et al.) than tested in the current studies.

We have partially answered these concerns by analyzing additional displacement speeds in the present work. This confirms the absence of coordination of breaths to strides. We agree that it will be important for future work to document a wider range of running speeds, longer bouts of gallop and other gaits, as well as more intense training and other context for running. Our revised Discussion touches upon these limitations and now makes clear that what is described in the present regimes may not apply to all running situations or individuals, even in mice.